# Digital signaling decouples activation probability and population heterogeneity

Ryan A Kellogg[1], Chengzhe Tian[2], Tomasz Lipniacki[3], Stephen R Quake[4], Savaş Tay[1,5]*

[1]Department of Biosystems Science and Engineering, Eidgenössische Technische Hochschule Zürich, Basel, Switzerland; [2]Niels Bohr Institute, University of Copenhagen, Copenhagen, Denmark; [3]Institute of Fundamental Technological Research, Polish Academy of Sciences, Warsaw, Poland; [4]Department of Bioengineering, Howard Hughes Medical Institute, Stanford University, Stanford, United States; [5]Institute for Molecular Engineering, University of Chicago, Chicago, United States

**Abstract** Digital signaling enhances robustness of cellular decisions in noisy environments, but it is unclear how digital systems transmit temporal information about a stimulus. To understand how temporal input information is encoded and decoded by the NF-κB system, we studied transcription factor dynamics and gene regulation under dose- and duration-modulated inflammatory inputs. Mathematical modeling predicted and microfluidic single-cell experiments confirmed that integral of the stimulus (or area, concentration × duration) controls the fraction of cells that activate NF-κB in the population. However, stimulus temporal profile determined NF-κB dynamics, cell-to-cell variability, and gene expression phenotype. A sustained, weak stimulation lead to heterogeneous activation and delayed timing that is transmitted to gene expression. In contrast, a transient, strong stimulus with the same area caused rapid and uniform dynamics. These results show that digital NF-κB signaling enables multidimensional control of cellular phenotype via input profile, allowing parallel and independent control of single-cell activation probability and population heterogeneity.

*For correspondence: savas.tay@bsse.ethz.ch

Competing interests: The authors declare that no competing interests exist.

## Introduction

Cells must make decisions in noisy environments and have to decrease the chance of an errant response. One way cells can reduce sensitivity to noise is through digital or switch-like activation, such that only sufficiently strong signal exceeds an internal threshold and initiates a response. Switch-like activation occurs through diverse mechanisms (*Shah and Sarkar, 2011*). For example, observations in *Xenopus* oocytes showed that the MAPK pathway converted graded progesterone input to digital output in p42 MAPK that determined oocyte maturation (*Petty et al., 1998*). Subsequently, similar observations were seen for the JNK pathway (*Bagowski and Ferrell, 2001*). The scaffolding protein Spe5 was found to mediate digital MAPK activation of mating in yeast (*Malleshaiah et al., 2010*). More recently, it was found that inflammasome signaling leads to all-or-none caspase1 activation that mediates apoptosis (*Liu et al., 2013*). Both amplitude (dose) and duration of input signals provide information that regulates cellular decisions. The duration of Epidermal Growth Factor (EGF) stimulation modulates ERK dynamics and controls differentiation (*Santos et al., 2007*; *von Kriegsheim et al., 2009*; *Ahmed et al., 2014*). Glucose sensing in plants showed that cells have gene regulatory network mechanisms to allow similar responses to a short, intense or sustained, moderate stimulus (*Fu et al., 2014*). Lymphocytes must precisely measure both antigen affinity and frequency to decide differentiation and proliferation (*Iezzi et al., 1998*; *Gottschalk et al., 2012*; *Miskov-Zivanov et al., 2013*). Although digital pathway activation allows robust

**eLife digest** Cells have communication systems called signaling pathways that enable them to detect and respond to changes in their surrounding environment. For example, in humans and other animals, a signaling pathway called NF-κB signaling is part of the immune system and regulates the inflammation that is caused by damage to cells, or by an invading microbe. Several signal molecules, including a protein called TNF—which is released by cells during an immune response—activate NF-κB signaling. However, the levels of TNF in the environment around a cell may fluctuate randomly even when there is no immune response. Therefore, the NF-κB pathway needs to be able to tell the difference between this 'noise' and a large increase in TNF associated with an immune response.

To get around this problem, many signaling pathways are activated in a switch-like manner so that only a strong signal that exceeds a particular threshold will lead to a response. These so called 'digital' responses help cells to filter out noise caused by random fluctuations in the amount of a signal molecule. NF-κB signaling responds to TNF in a digital manner, but it is not clear how information about the length of the signal can influence the degree to which NF-κB signaling is activated.

Kellogg et al. used a combination of mathematical modeling and microscopy techniques to study the activation of NF-κB signaling in mouse cells. The study shows that a molecule called LPS—which is produced by microbes known as bacteria—can also switch on the signaling pathway in a digital manner, but in a different way to TNF. In a population of cells, the fraction that activate NF-κB signaling in response to LPS or another signal is determined by the level of the signal (also known as its 'concentration') multiplied by the signal's duration. This is known as the signal's 'area'.

On the other hand, the way that these cells respond to the activation of NF-κB signaling depends on the nature of the activity produced by the signal pathway. For example, a short but strong burst of LPS signal leads to rapid and uniform responses in the cells. A weaker but longer lasting signaling activity leads to slower, more varied responses in cells.

These findings reveal that such switch-like, digital responses do more than just filter out noisy signals. They can also integrate information about the timing and intensity of the signal to independently control different aspects of cell responses. The next challenge will be to extend this understanding to more complex scenarios, such as when signals contain several types of molecules at the same time.

cellular decision across a wide range of systems, it is not clear how digital signaling impacts processing of dose and duration information.

NF-κB is a critical regulator of phenotype in immunity and disease (*Hayden and Ghosh, 2008*) and responds digitally to Tumor Necrosis Factor (TNF) stimulation (*Tay et al., 2010*; *Turner et al., 2010*). NF-κB activation occurs for a multitude of cell stress and inflammatory signals that converge on the IKK (IκB Kinase) signaling hub, which induces degradation of the cytoplasmic inhibitor IκB and liberates NF-κB to enter the nucleus and regulate gene expression (*Hayden and Ghosh, 2008*). Multi-layered negative and positive feedback lead to complex pathway dynamics including oscillations (*Hoffmann et al., 2002*; *Nelson et al., 2004*; *Tay et al., 2010*; *Kellogg and Tay, 2015*). Although it is not fully resolved how NF-κB coordinates gene and phenotype regulation, it is known that dynamic NF-κB activation is involved in input–output specificity and information transmission (*Werner et al., 2005*; *Ashall et al., 2009*; *Behar and Hoffmann, 2013*; *Selimkhanov et al., 2014*). The core IκB-NF-κB regulatory module is well-studied and appears largely consistent across multiple stimulation contexts (*Hoffmann et al., 2002*; *Nelson et al., 2004*; *Tay et al., 2010*; *Hughey et al., 2014*); however, the role of module upstream of IKK activation including receptor-ligand binding and adaptor protein assembly in input-encoding remains unclear.

To probe how diverse IKK-upstream signaling architectures impact NF-κB processing of pathogen- and host-associated inflammatory inputs, we used microfluidic cell culture to precisely modulate dose and duration of LPS and TNF stimuli and measured NF-κB dynamics using live cell imaging (*Figure 1*) (*Junkin and Tay, 2014*; *Kellogg et al., 2014*). We found that lipopolysaccharide (LPS) induces NF-κB activation in a digital way where cells respond in an all-or-none fashion, but in a distinct manner from TNF, with greater ultrasensitivity and pronounced input-dependent activation delay. Computational

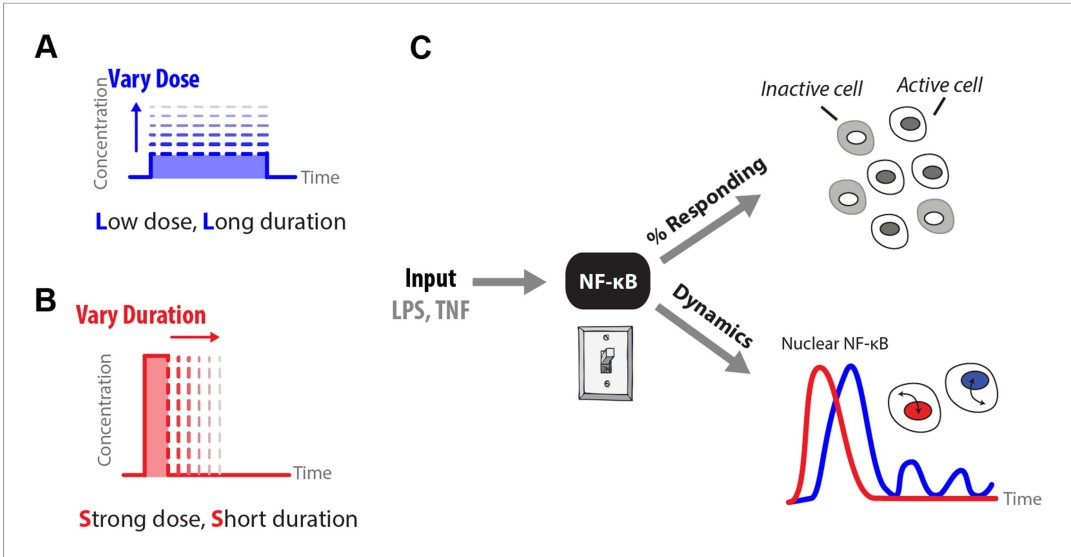

**Figure 1**. How does input profile determine digital signaling response? Since the amplitude and time profile of input signals depends on biological context, such as distance to an infection site or pathogen loading, we use microfluidics to manipulate dose (**A**) and duration (**B**) of LPS and TNF input signals, which induces digital activation of NF-κB. (**C**) Switch-like digital NF-κB responses are analyzed in terms of fraction of cells that activate in the population and heterogeneity in the dynamic responses in activating cells.

modeling predicted and experiments confirmed that LPS integral over the stimulus or 'area' (concentration × duration) controls the percentage of cells that activate in the population. Importantly, dynamics of NF-κB activation depend on input temporal profile, so that a long duration, low-dose (LL) signal induces delayed, heterogeneous activation timing in the population while a short duration, strong amplitude (SS) signal with the same area causes rapid activation without cell-to-cell timing variability (*Figure 1*). These results reveal a function for digital signaling beyond simple noise filtering: digital activation controls fate along a two dimensional space by allowing an input signal to independently control the population response (percentage of responding cells) and single-cell response (transcription factor dynamics and gene expression phenotype) though modulation of signal area and shape.

## Results

### NF-κB switch dynamics distinguish pathogen (LPS) and host (TNF) signals

To initially evaluate the behavior of the LPS/NF-κB pathway, we stimulated 3T3 NF-κB reporter cells (*Lee et al., 2009*; *Tay et al., 2010*) with different concentrations of LPS in a microfluidic system (*Gómez-Sjöberg et al., 2007*; *Kellogg et al., 2014*) and performed time-lapse live microscopy to record NF-κB nucleus-cytoplasm translocation over time (*Figure 2A*). Each experimental condition is measured in duplicate chambers on the chip. We found that LPS-exposed cells activated NF-κB in an ultrasensitive, digital fashion. The population consisted of cells either responding or ignoring the LPS input, with the percentage of responding cells in the population scaling with LPS concentration, from 5% at 0.25 ng/ml to 100% at 500 ng/ml (*Figure 2B,D*). Amplitude is highly variable across doses. Median amplitude increases gradually with dose though this change is statistically less significant than change in response time (*Figure 2—figure supplement 2*). NF-κB dynamics in activating cells showed small oscillations beyond the first peak. When ligand is flowed continuously through the chamber to replace ligand loss due to cellular internalization, oscillations sustain for the duration of the stimulus (*Figure 2—figure supplement 1A*). Under low intensity LPS stimulation, most cells did not respond (*Figure 2C,D*). This was a similar effect as previously observed under TNF (*Tay et al., 2010*; *Turner et al., 2010*). While both LPS and TNF are digital in preserving first peak area, the TNF-induced NF-κB initial peak becomes flatter and wider with increasing response time but unchanging onset time for

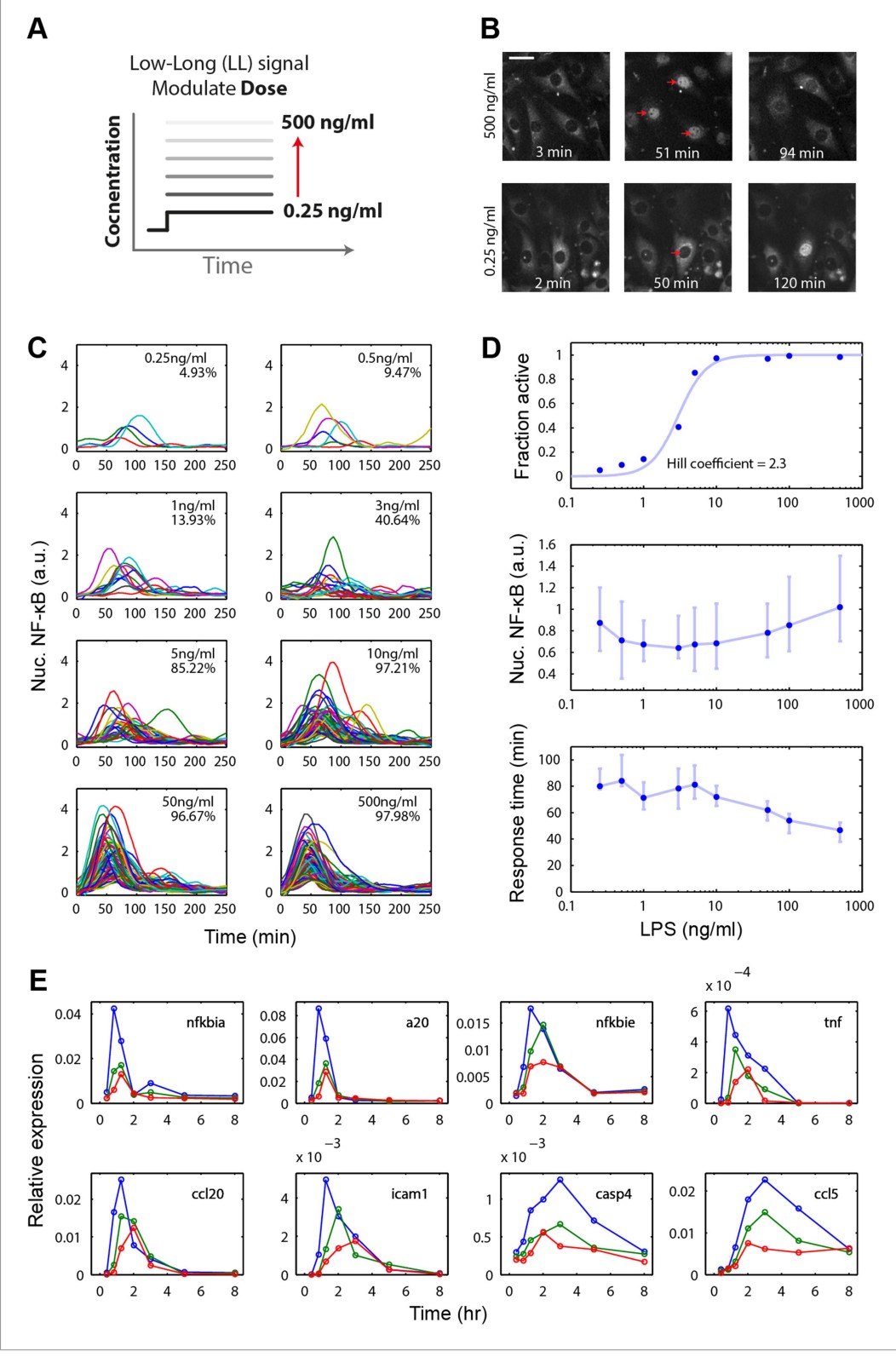

**Figure 2**. Digital, time-delayed NF-κB activation under varied LPS dose stimulation. (**A**) Cells process pathogen signal dose and duration to dynamically activate NF-κB, which induces gene expression and coordinates the innate immune response. We test the role of pathogen load by varying LPS concentration from 0.25 ng/ml to 500 ng/ml using microfluidic cell culture. (**B**) Time series images of NF-κB activation following LPS treatment. Top row: high LPS
*Figure 2. continued on next page*

*Figure 2. Continued*

dose causes nearly 100% of cells to respond synchronously. Bottom row: at low LPS concentration, less than 5% of cells respond and initiate NF-κB activation with variable, delayed timing. The cells respond digitally, with nearly all cytoplasmic NF-κB moving into the nucleus. The response amplitude (indicated by peak intensity of nuclear p65-dsRed fluorescence) depends on the initial NF-κB abundance in the nucleus and exhibits high variability across doses. (C) Trajectories of NF-κB activation (intensity of nuclear p65-dsRed) tracked in single cells over time for LPS doses ranging from 500 to 0.25 ng/ml. As the LPS dose decreases, response timing becomes delayed and variable, and the percent of responding cells in the population drops. (D) Across the LPS doses tested: top panel, dose-response curve of the fraction of active cells (plotted is the mean of two duplicate cell chambers in the chip for each condition), middle panel, the intensity of nuclear NF-κB at the peak of the response, and lower panel, time until the peak of the response. Peak nuclear NF-κB amplitude is highly variable across doses. The dose response shows a sharp drop in fraction of active cells between 1 and 5 ng/ml concentration, indicating that the activation threshold is within this range for most cells. With lower dose, the response time increases in both median duration and variability. In middle and lower panels, data points and error bars represent median and interquartile range, respectively. (E) NF-κB dependent gene expression dynamics under varied LPS concentrations (blue: 500 ng/ml, green: 100 ng/ml, red: 50 ng/ml). With lower LPS concentration, several genes show delayed induction. TNF dose-modulated expression of the same genes can be found in Figure 3 of *Tay et al. (2010)* and Figure 2A of *Pękalski et al. (2013)*.

The following figure supplements are available for figure 2:

**Figure supplement 1**. Digital, time-delayed NF-κB activation under continuous LPS stimulation.

**Figure supplement 2**. Statistical analysis for NF-κB peak amplitude and timing measurements under LPS dose modulation (corresponding to *Figure 2D*).

decreasing input dose, while LPS experiences greater dose-dependent onset delay and timing variability and maintains a consistent peak shape (*Figure 2C* and *Figure 2—figure supplement 1B*). The dose-response curve for LPS was steeper than that for TNF (fitted to Hill dynamics reveals Hill coefficients of 2.3 and 1.5, respectively) (*Figure 2C*, *Figure 2—figure supplement 1C*, *Figure 5—figure supplement 1B*). These results indicate that the LPS pathway activates in a switch-like manner, with increasing fraction of cells in the population responding as dose is increased, but with distinct activation dynamics compared to TNF input.

## Response timing and single-cell heterogeneity depends on stimulus intensity

We next analyzed dynamics of the NF-κB response in those cells that activate. Notably, there were differences in the timing of the response for high versus low dose. High-dose long duration input caused a rapid response with the response peak occurring at approximately 35 min after stimulation. In contrast, low-dose long duration (LL) input led to a pronounced statistically significant delay in the response (*Figure 2C,D* and *Figure 2—figure supplement 2*). At lowest doses, the median delay until the peak of the response exceeded 80 min with heterogeneity in the response timing between cells (*Figure 2C,D*).

We next asked whether this delayed response impacted LPS and NF-κB-mediated gene expression. We explored how the increase in delay for 500 ng/ml LPS versus 50 ng/ml LPS impacted gene regulation. Notably, gene expression of early and intermediate genes exhibited a dose-dependent delay (*Figure 2E*). The extent and magnitude of the dose-dependent delay and heterogeneity differs from TNF stimulation of the same cell type (*Tay et al., 2010*). While decreasing TNF dose altered the response slope, LPS response maintained a stereotypical peak shape that shifts later in time with lower dose (*Figure 2E*). Delayed gene expression observed under LPS stimulation contrasts with TNF-α input that does not induce delayed induction of these genes (*Tay et al., 2010*; *Pękalski et al., 2013*). For early genes IκBα and A20, gene expression peak is shifted from 30 min to 1 hr after stimulation. IκBε expression shifts from maximum expression at 1 hr–2 hr and from 30 min to 2 hr for TNF mRNA under LPS input. Intermediate genes *Ccl2* and *Icam* shift expression peaks from

1 hr to 2 hr and from 1 hr to 3 hr (*Figure 2E*), respectively. Late genes *Ccl5* and *Casp4* do not reflect the delayed NF-κB activation due to slower induction kinetics. Both NF-κB dynamics in microfluidics and mRNA responses in tissue culture may be affected by autocrine signaling loops (*Pękalski et al., 2013*). Overall, these results indicate that LPS induces digital NF-κB activation with an input dose-dependent delay that carries through to gene expression dynamics.

## Cooperative IKK activation underlies dose-dependent response delay

To study how various pathway components upstream of IKK influence input information transfer to NF-κB, we developed a model of LPS-induced NF-κB switch activation. LPS activates NF-κB by TLR4 engagement via CD14, leading to TLR4 dimerization. TLR4 dimers recruit MyD88, IRAK2/4, and other adaptor proteins leading to clustering and higher order assembly of Myddosome and TRAF6 lattice structures, which cooperatively activates IKK (*Yin et al., 2009*; *Lin et al., 2010*; *Zanoni et al., 2011*). Following IKK activation, nuclear NF-κB induces expression of IκBα, which negatively regulates NF-κB and IKK, respectively (*Hayden and Ghosh, 2008*). Experimental IκBα expression kinetics were similar for LPS and TNF, despite induction delay under LPS (*Figure 2E*). Multiple efforts have modeled NF-κB pathway dynamics under TNF stimulation (*Hoffmann et al., 2002*; *Lipniacki et al., 2007*; *Ashall et al., 2009*; *Paszek et al., 2010*; *Tay et al., 2010*; *Pękalski et al., 2013*). Extrinsic noise including variation in receptor-level and pathway components contributes to cell-to-cell heterogeneity in cell sensitivity and response dynamics (*Snijder et al., 2009*; *Tay et al., 2010*). We based our mathematical model on the core IKK-NF-κB regulatory module (*Tay et al., 2010*), which has been extensively validated experimentally.

To extend the NF-κB core model for LPS, we added species for LPS, TLR4, and TRAF6 (Appendix 1) (*Figure 3A*). To introduce variability in the model, we allowed fluctuation in the number of TLR4 receptor molecules between cells. TLR4 is expressed at relatively low level compared to CD14 and furthermore varies significantly between cells in the population (*Zanoni et al., 2011*). To account for cooperative activation due to Myddosome assembly and TRAF6 lattice formation, we model IKK phosphorylation by TRAF6 using Hill kinetics. The model reproduced the observed LPS induced NF-κB dynamics in single cells for different LPS doses (*Figure 3B–D* and *Figure 3—figure supplement 1*), though showed more dose-dependent first peak amplitude variation than observed in experiments. The distinct feature of the proposed LPS model is the Myddosome formation leading to cooperative activation of IKK (coopertivity coefficient = 4, note this value is distinct from the slope of the active cell fraction response curve), which simultaneously assures delay in activation observed experimentally for low doses, and steeper response curve than in the case of TNF stimulation (*Figure 3* and *Figure 2—figure supplement 1*).

## Input duration controls activation probability independent of response heterogeneity

Cellular environments like infected tissue encode information in both amplitude and duration of input signals (*Gottschalk et al., 2012*; *Fu et al., 2014*). To understand how pathogen input signal duration and input integral or area (concentration × duration) impact digital NF-κB signaling, we performed a simulated screen across a large range of LPS concentration and duration combinations using our model. We first observed that just as LPS concentration modulates fraction of active cells so does duration. Simulations keeping concentration high and changing input duration on a short, sub-minute timescale altered the percent of activating cells (*Figure 4*). Nearly, all cells respond for durations exceeding 1 min at 500 ng/ml. Notably, in contrast to changing concentration under constant long duration, which introduces timing delay and heterogeneity (*Figures 2C, 3D*), changing sub-minute duration of a high-amplitude signal in simulation controlled fraction of active cells while maintaining uniformly timed, rapid NF-κB responses (*Figure 4B,D* and *Figure 4—figure supplement 1*).

We experimentally provided pulsed LPS at 500 ng/ml for sub-minute durations using microfluidic cell culture (*Figure 5A*). In agreement with simulation predictions, we found that precisely controlled stimulus duration regulated the activation of cells in the population in a strongly all-or-none manner, with 1- to 40-s duration LPS exposure (500 ng/ml) activating ~3–88% of the population (*Figure 5B* and Supplementary *Videos 1, 2*). Short duration (i.e., 1 s) stimulation, mimicking very brief exposure to bacteria, activated a small percentage of cells in the population. Moreover, under short duration, strong amplitude (SS) input, responses were fast and uniform (*Figure 5B*, top row) in contrast to low,

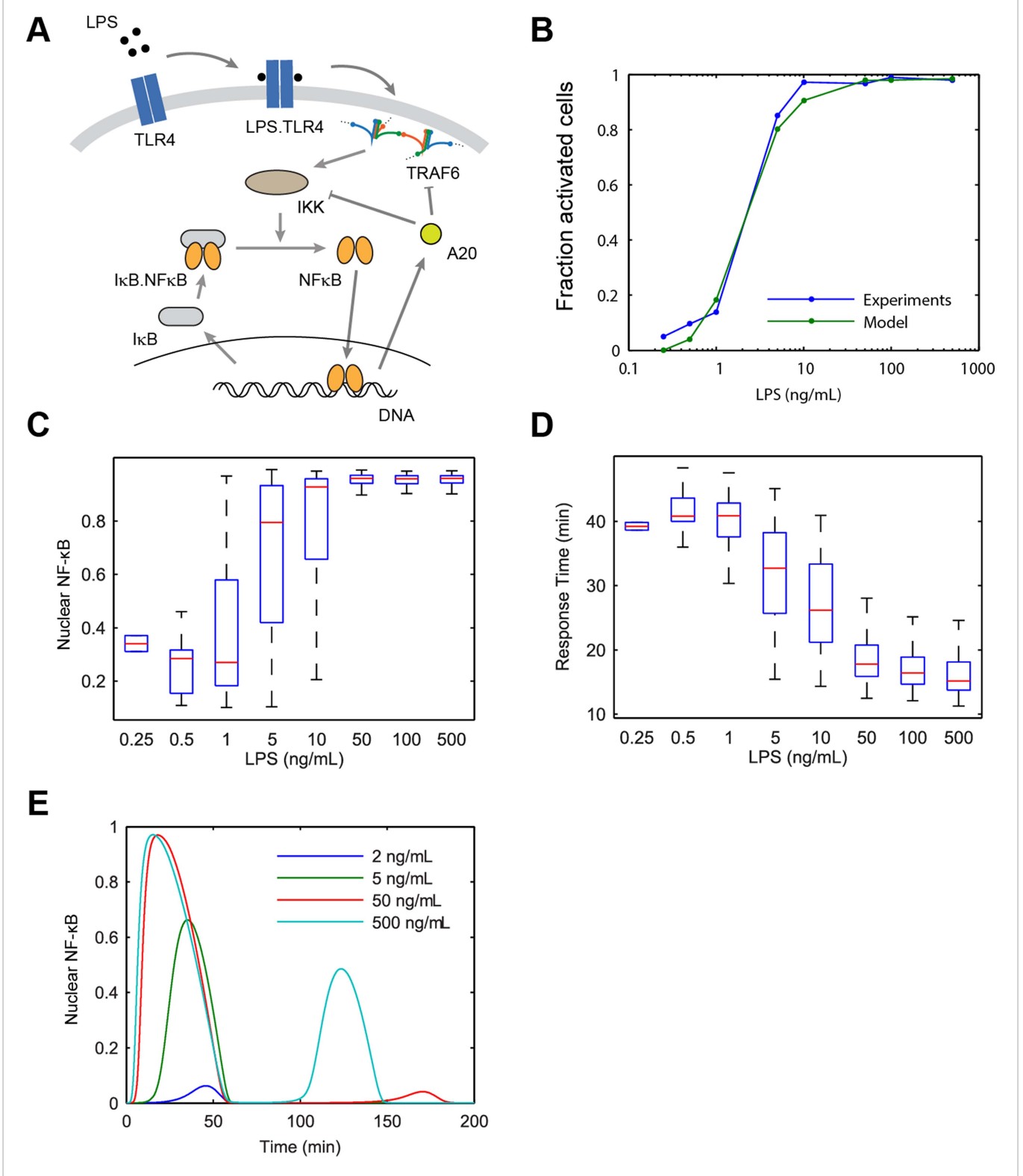

**Figure 3**. Model scheme and simulation of LPS dose modulation. (**A**) The scheme of the model. LPS binds TLR4 leading to TRAF6 activation, which cooperatively activates IKK. Active IKK induces IκB degradation, which allows NF-κB to enter the nucleus and upregulate expression of IκB and A20. New IκB sequesters NF-κB in the cytoplasm and A20 inhibits upstream pathway activation by IKK and TRAF6. (**B**) Simulated versus experimental LPS dose response. (**C**) NF-κB peak intensities (expressed as proportion of total NF-κB molecules in the nucleus). (**D**) NF-κB response time as function of LPS dose.

*Figure 3. continued on next page*

*Figure 3. Continued*

(**E**) Sample simulated curves of nuclear NF-κB fractions under LPS treatment. In box and whisker plots (**C**, **D**), the central red line is the median, the edges of the box are the 25th and 75th percentiles, and the whiskers extend to the most extreme data points.
The following figure supplements are available for figure 3:

**Figure supplement 1**. Simulated NF-κB trajectories for various doses of LPS treatment.

**Figure supplement 2**. Modelling predictions for the LPS pathway.

long (LL) stimulation that led to delayed, variable responses (*Figure 2C*). For example, 3–5% activation occurred for both a 1-s short pulse at 500 ng/ml LPS (SS signal) and a 0.25 ng/ml constant input signal (LL signal) (*Figure 5B*, *Figure 2D*). However, modulating duration of the SS signal from 1 s to 40 s (activating 3.3% and 87.5% the population, respectively) changes median response timing by less than 2 min (*Figure 5B,C*). Statistical analysis indicates no significant difference in response time under varied pulse durations (*Figure 5—figure supplement 2*). In contrast, modulating concentration of the

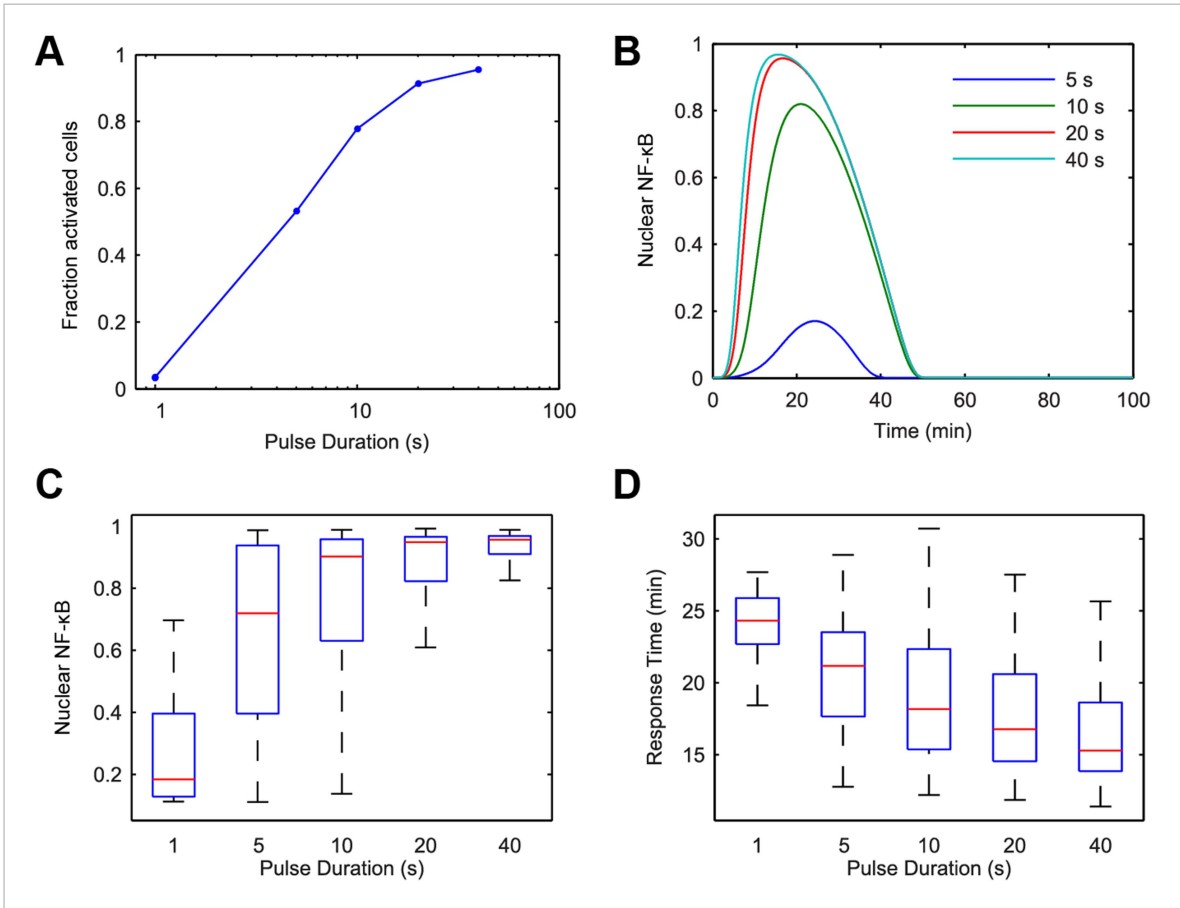

**Figure 4**. Model simulation predicts that stimulus duration controls fraction of activating cells and response timing variability. (**A**) Simulated fractions of activated cells under increasing duration of 500 ng/ml LPS pulse. (**B**) Sample simulated curves of NF-κB under 5- to 40-s duration LPS (500 ng/ml) pulse. (**C**) Distributions of nuclear NF-κB amplitude and (**D**) response times of activating cells under various durations of LPS treatment. In box and whisker plots (**C**, **D**), the central red line is the median, the edges of the box are the 25th and 75th percentiles, and the whiskers extend to the most extreme data points.
The following figure supplement is available for figure 4:

**Figure supplement 1**. (**A–F**) Simulated NF-κB single-cell trajectories with randomly sampled numbers of TLR4 and NF-κB for 1- to 60-s durations of LPS exposure.

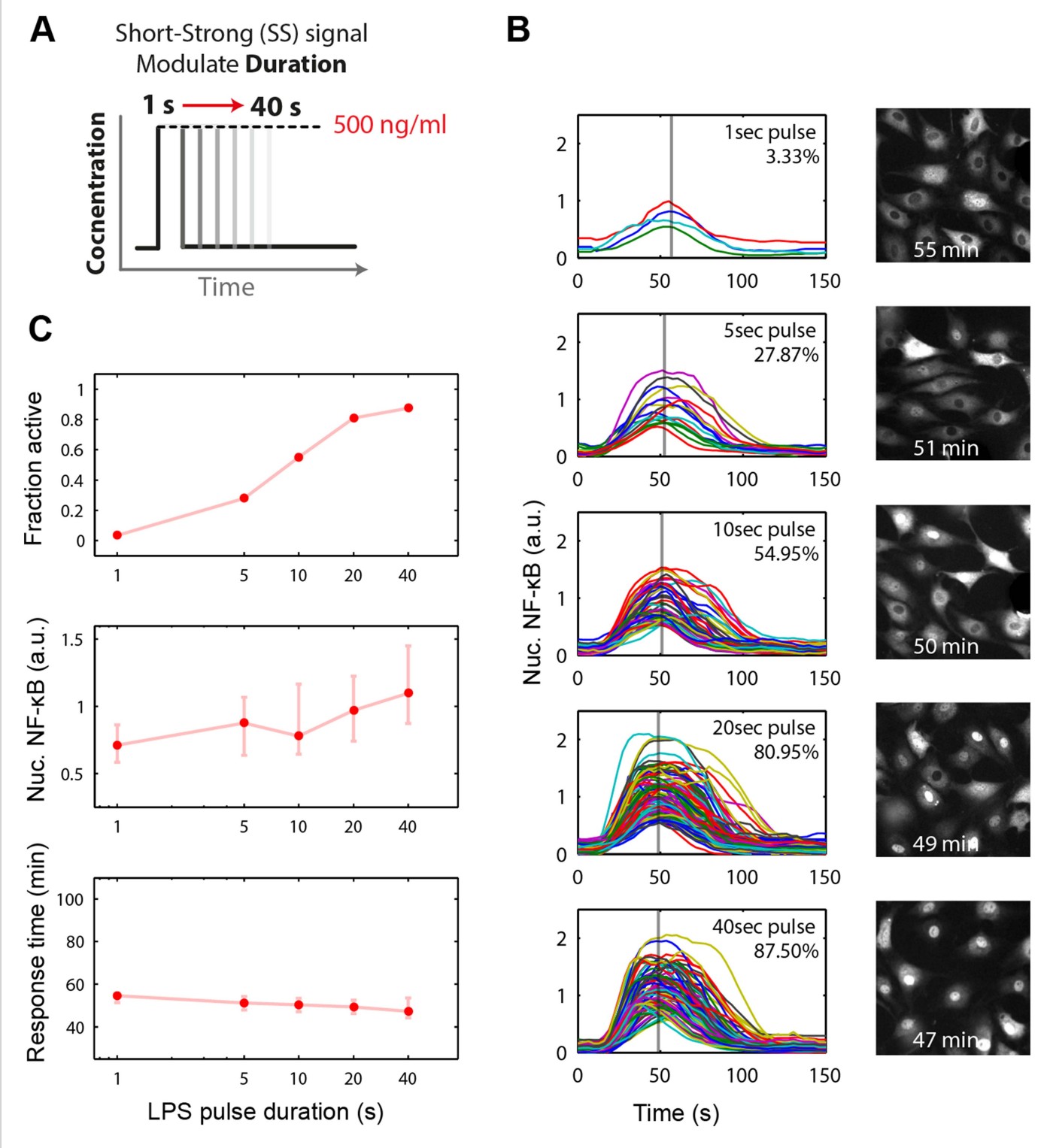

**Figure 5**. Short duration LPS pulse stimulation modulates responding cell fraction and fast, uniformly timed response. (**A**) LPS duration is manipulated using microfluidic cell culture in the range of 1–40 s. Dose is held constant at 500 ng/ml. (**B**) Single-cell NF-κB trajectories for 1- to 40-s duration LPS pulse stimulation. Short pulse LPS reduces variation in timing in the start of NF-κB activation. (**C**) Top panel: fraction of active cells as a function of LPS pulse duration. (Plotted points are the mean of two duplicate chambers in chip.) Middle panel: NF-κB nuclear response intensity as a function of LPS pulse duration. Lower panel: time of the NF-κB response peak as a function of LPS pulse duration. Timing variability is dramatically reduced under short-pulsed

*Figure 5. continued on next page*

*Figure 5. Continued*

stimulation (see **Figure 2D** for comparison to constant stimulation). In middle and lower panels, data points and error bars represent median and interquartile range, respectively.

The following figure supplements are available for figure 5:

**Figure supplement 1**. NF-κB dynamics under pulsed stimulation with TNF at 10 ng/ml concentration.

**Figure supplement 2**. Statistical analysis for NF-κB peak amplitude and timing measurements under LPS duration modulation (corresponding to *Figure 5B*).

**Figure supplement 3**. Statistical analysis for NF-κB peak amplitude and timing measurements under TNF (10 ng/ml) duration modulation (corresponding to *Figure 5—figure supplement 1*).

LL signal from 0.25 to 500 ng/ml (activating 4.9% and 98% of the population, respectively) changes the response time more than 35 min (reduced from 80 to 43 min). Moreover, while the variability in the response time scales with dose under LL stimulation, timing variability remains low under duration-modulated SS stimulation (**Figures 4D, 5C**). From an immunological perspective, this experiment indicates that brief but high pathogen load leads to uniform and strong NF-κB response in the population, while chronic low-grade pathogen exposure leads to population variability and delay.

We next changed the stimulus type to TNF instead of LPS. In vivo, TNF is secreted from immune cells that come in contact with pathogenic signals like LPS. Again, we observed the phenomena that the fraction of active cells changed while the response timing did not (**Figure 5—figure supplement 1**), though amplitude is more significantly affected (**Figure 5—figure supplement 3**). Together, these results indicated that SS input achieves control over the fraction of cells activating without affecting

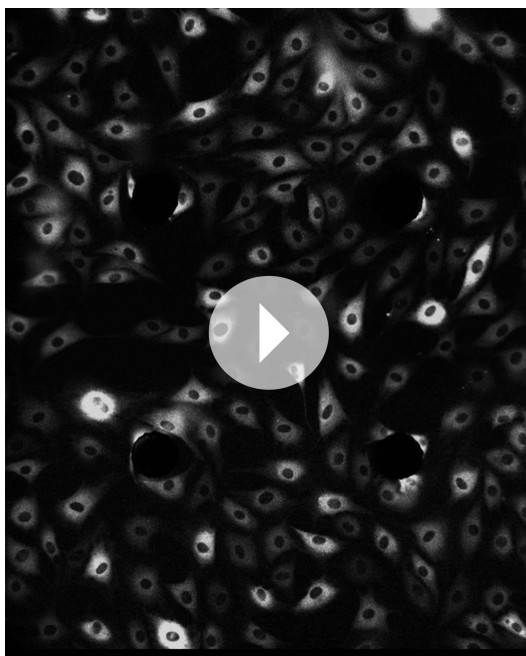

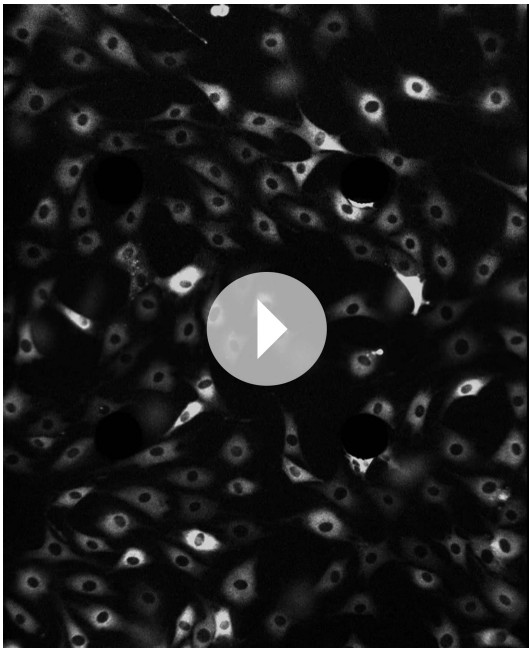

**Video 1.** Digital NF-κB response unders 1 second duration LPS exposure (500 ng/ml). This video shows fibroblast cells expressing NF-κB p65-dsRed responding in a digital fashion to brief (1-s duration) stimulus in a microfluidic chamber. Only 3–4% of cells show a response.

**Video 2.** Digital NF-κB response under 10 second duration LPS exposure (500 ng/ml). This video shows fibroblast cells expressing NF-κB p65-dsRed responding in a digital fashion to 20-s duration stimulus in a microfluidic chamber, activating more than half (~55%) of cells in the population.

the dynamics in the response. Therefore, duration sensing allows control of percentage of cells that produce a response without affecting response timing or heterogeneity. This contrasts to amplitude (concentration) sensing, where response dynamics in activating cells differs for high versus low amplitude. Since NF-κB dynamics influence gene expression, duration modulation to control percent population activation is therefore a strategy to achieve more homogeneous gene expression and phenotype outcomes between cells.

## Integral of stimulus determines fraction of active cells in the population

We sought to fully characterize the relationship between signal amplitude and duration in NF-κB switch activation. Since modulating either amplitude or duration was able to change the percentage of activating cells, we hypothesized that the fraction of activation may depend on the integral of the input (concentration × duration). Indeed, mathematical analytical analysis suggested that percent activating cells should scale with the input area (Appendix 1).

To validate our mathematical analysis and clarify how digital activation integrates stimulus dose and duration, we performed simulations. Each simulation series fixed the LPS stimulus dose and varied duration from 1 to 500 s. The output of these simulations as a function of stimulus duration shows multiple dose-response curves that do not coincide, indicating that duration is not the only predictor of switching probability or fraction of active cells (*Figure 6A*). However, when instead plotted as a function of stimulus area (concentration × duration), all simulation series closely coincide, indicating that stimulus area clearly determines the percentage of cells that activate in the population (*Figure 6A*).

To illustrate further the relationship between stimulus area and percentage of active cells, we plotted for each simulation dose the minimum duration needed to achieve 10%, 50%, and 90% activation (*Figure 6B*). This analysis revealed a reciprocal relationship between dose and duration in NF-κB switch activation (high dose requires less duration to achieve activation and vice versa). Simulations therefore supported analytical derivation of an 'Area Rule' in which concentration × duration determines the percentage of cells that activate in the population for a given stimulus.

Importantly, for concentrations that achieve less that 100% activation, increasing the duration infinitely will not further increase the active fraction (*Figure 6B*). Once duration is sufficiently long to activate the maximal potential cell fraction for a given dose, further increases in area by lengthening duration do not further increase percentage of active cells, indicating a limitation in the Area Rule.

We next simulated whether the Area Rule holds for fluctuating signals. When we compared a constant input signal to square wave input signals, with one square wave that 'starts high' and another that 'starts low', simulation revealed an equal percentage of activating cells when the two opposing square waves have equal area (i.e., the duration is a multiple of the square wave period) (*Figure 6C*). Further, the fraction of active cells matched that for a constant input signal with the same area (*Figure 6C*). Performing identical simulations using a model of TNF-induced NF-κB activation (*Tay et al., 2010*), we found that the Area Rule held also for the TNF network (*Figure 6—figure supplement 1*).

We found that in both the LPS amplitude-modulated and duration-modulated microfluidic experiments, stimulus area is an accurate predictor of fraction of active cells in the population (*Figure 7A*), as predicted by the model simulations. To experimentally verify that the integral over time of the input signal determines the fraction of activating cells, we performed microfluidic experiments varying temporal profile while maintaining the same integrated area. We observed 500 ng/ml LPS pulsed for 10 s activated approximately half the population (*Figure 5C*). Therefore, we tested two additional input profiles having the same area (5000 ng ml$^{-1}$*s): 50 ng/ml for 100 s and 100 ng/ml for 50 s. In agreement with model prediction, each of these conditions also activates a similar fraction of the population (51% and 54%, respectively) (*Figure 7C*). Together, experimental findings, simulations, and mathematical analysis demonstrate how cells integrate amplitude and duration of input signals in switch-like pathway activation. Stimulus integral (or area) determines the effective 'probability' that a given cell activates NF-κB (based on the percentage that activate in the population). These results indicate that the pathogen load (i.e., LPS dose) and duration of exposure (i.e., LPS pulse duration) are integrated by NF-κB system and together determine the population response.

## Amplitude and duration information transfer via digital NF-κB activation

We showed that modulating stimulus amplitude altered response dynamics by changing the amount of activation delay. In contrast, modulating stimulus duration did not affect activation delay but

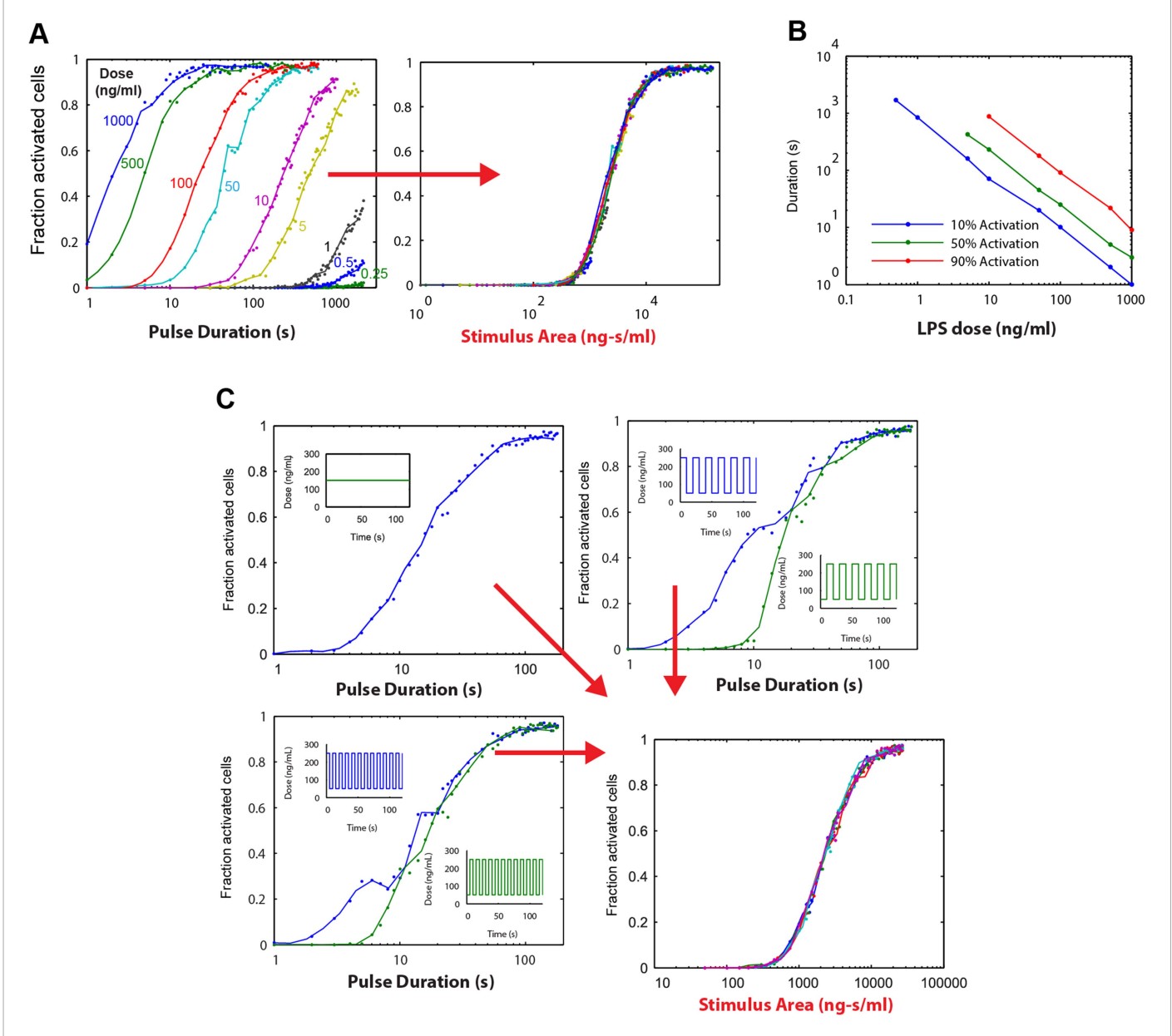

**Figure 6**. Simulations demonstrating an 'Area Rule', that is, the relationship between LPS stimulus area and fraction of active cells. (**A**) The simulated fractions of activated cells for pulsed inputs of LPS with various doses and durations. Each fraction is estimated by 500 independent simulations. When the points are plotted as a function of stimulus area (rather than duration), all points fall on the same curve, indicating that stimulus area tightly controls the fraction of active cells. (**B**) The minimal duration for certain fractions of activation as a function of dose. The minimal duration is determined by searching for the first tested time point where the estimated fractions of activation are above the threshold. The doses for which the threshold level cannot be achieved are not shown in the figure. Blue: 10% activation, green: 50% activation, red: 90% activation. (**C**) Further verification of the relationship between stimulus area and active cell fraction using square wave input profiles. Equal area input was generated using either a single pulse (top left), square wave with 10-s period (lower left), or square wave with 20-s period (top right). Regardless of input shape, all simulated points fall on the same curve when plotted as a function of stimulus area. For square wave inputs, one input begins high (blue) while another input (green) begins low. Note that the curves intersect for durations 10 s, 20 s, 30 s (or 20 s, 40 s, 0 s, … for the input with 20-s period) when area under the two signals is the same.

The following figure supplement is available for figure 6:

**Figure supplement 1**. Stimulus area simulation using a TNF model (*Tay et al., 2010*) revealed that the 'Area Rule' holds also for TNF.

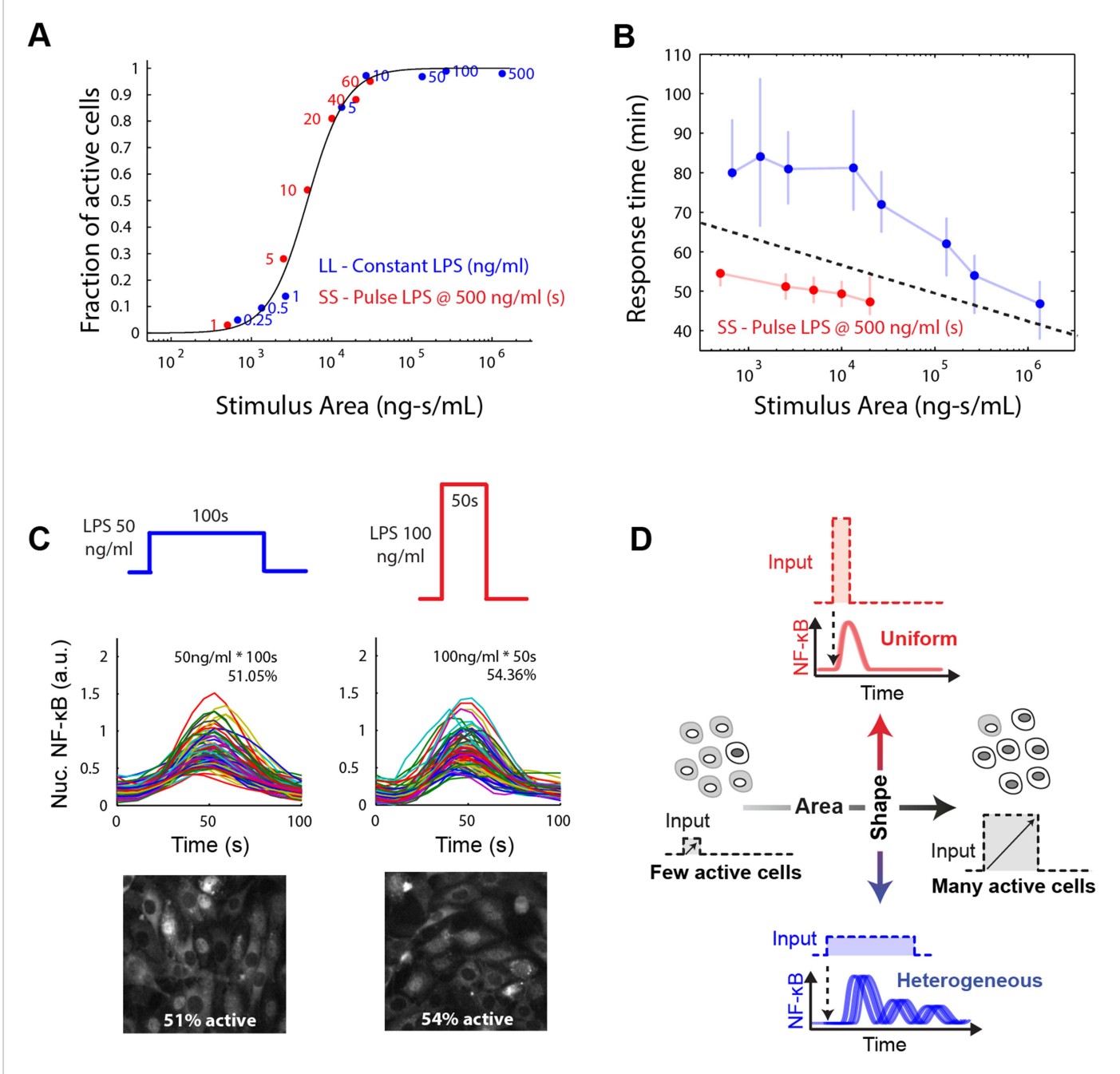

**Figure 7.** Stimulus area determines NF-κB population response. (**A**) Stimulus area determines fraction of active cells. The experimentally tested dose and duration inputs fall on the same hill-like activation curve when plotted as a function of stimulus area, as predicted by model simulations, indicating that total integrated ligand concentration (stimulus area) controls the probability of cell activation. These results show that pathogen load (i.e., LPS dose) and duration of exposure (i.e., LPS pulse duration) are integrated by NF-κB system and together determine the population response. (**B**) Response time discriminates between sustained, low intensity (blue) and transient, high intensity (red) stimulus. Data points and error bars represent median and interquartile range, respectively. (**C**) Experimental verification that stimulus integral over time determines the fraction of active cells. A pulse of either LPS 50 ng/ml for 100-s duration or 100 ng/ml for 50-s duration generated approximately the same responding cell percentage as a, 51% and 54% for the two inputs, respectively. (**D**) Input profile controls digital responses along two axes: integral over stimulus (area) controls the fraction of activated cells in the population. Input temporal profile (shape) controls dynamic heterogeneity in responding cells. In (**B**), data points and error bars represent median and interquartile range.

The following figure supplement is available for figure 7:

**Figure supplement 1.** LPS response heterogeneity under pulsed and continuous input.

changed the population variability. These findings indicate tradeoffs in dose versus duration sensing. Duration sensing allows for controlling only the population response while not affecting the single-cell response, so that it is possible to achieve homogeneous dynamics and uniform phenotype in a desired proportion of cells. In contrast, it may be useful to transmit information that can instruct different dynamics and phenotype, which is achieved by modulating dose. While dose information is transmitted through the NF-κB digital response, duration information is lost at the single-cell level.

However, transmitting information using only dose modulation necessarily changes the percentage of cells in the population that respond. In physiological settings, it may be desirable to transmit information without affecting population response, that is, for a signal to affect response dynamics in activating cells without impacting the proportion that activate. To achieve this requires modulating both dose and duration to maintain input area, leading to a shift in input temporal profile from a SS to a LL signal. Cells distinguish an SS versus LL signal profile based on NF-κB and gene expression dynamics. We show that an intense, brief (SS) signal induces distinct dynamics than a weak, sustained (LL) signal, but the percentage of cells responding is the same in both cases (*Figure 2C*, *Figure 5B*). Response timing and intensity are dynamic features that provide information for discriminating input temporal profile. Indeed, plotting response delay as a function of stimulus area shows that SS and LL signals can be distinguished on the basis of response delay (*Figure 7B*). Cells can discern the category of the signal (whether SS or LL) for a given input area based on whether the response time falls above or below a separation line (*Figure 7B*). Modulating input amplitude associated with higher timing variability than input duration modulation for controlling the fraction of active cells (*Figure 7—figure supplement 1A*). Because response amplitude exhibits high variability between cells, amplitude alone does not provide sufficient information to discriminate an SS versus LL signal (*Figure 7—figure supplement 1B*). Physiological cues are in fact commonly transmitted by changing from an SS to an LL input profile (*Iezzi et al., 1998*; *Fu et al., 2014*). Biological systems therefore appear to take advantage of the unique ability of digital signaling to separate control of population and single-cell dynamics, by modulating input area to determine the proportion of cells that activate in the population and input shape to instruct to determine phenotype outcomes in the activating subset of cells (*Figure 7*).

## Discussion

This study asked how stimulus amplitude and duration determine NF-κB digital activation. Modeling and experiments showed that NF-κB activation is achieved by integrating the input: stimulus integral or area (concentration × duration) controlled the percentage of cells that activated for both a 'foreign' pathogen signal LPS and a 'self' immune signal TNF (population response). However, switch dynamics and gene expression phenotype varied depending on the input dose (single-cell response), with rapid homogeneous responses at high dose and delayed heterogeneous responses at low dose. Dynamics of transcription factor activation determine the timing and specificity of gene expression and phenotype responses (*Werner et al., 2005*; *Kobayashi et al., 2009*; *Purvis et al., 2012*). Therefore, intercellular signaling systems may achieve distinct phenotype outcomes by controlling the input shape or temporal profile (whether SS or LL), while input area determines percentage of cells that respond (*Figure 7B*). Greater heterogeneity with decreasing dose and decreased heterogeneity under short duration input is measured by coefficient of variation (*Figure 7—figure supplement 1*).

In lymphocyte signaling, T- and B-cells' cell fate depends on both antigen quality (affinity) and quantity (amount of presented antigen). Antigen quality is encoded in the duration of receptor-antigen contact, with characteristic interaction times on the order of seconds (*Altan-Bonnet and Germain, 2005*; *Gottschalk et al., 2012*; *Miskov-Zivanov et al., 2013*). T- and B-cell receptor binding with antigen-MHC triggers digital activation and cell fate control via NF-κB (*Kingeter et al., 2010*; *Oh and Ghosh, 2013*; *Gerondakis et al., 2014*; *Shinohara et al., 2014*). A reciprocal relationship is observed between antigen quality and quantity in lymphocyte activation: Higher antigen affinity requires lower dose of antigen to trigger T-cell proliferation, and inversely, lower affinity requires higher dose (*Gottschalk et al., 2012*). Moreover, an intense, transient compared to a weak, sustained signal induces positive versus negative selection of naive thymic T cells (*Iezzi et al., 1998*) and T helper cell differentiation into alternatively CD4 or CD8 status (*Adachi and Iwata, 2002*). Therefore, analogous to our findings, while a combination of antigen dose and contact duration determines the probability of activation, input profile determined by relationship between antigen quality and quantity decides the phenotypic outcome of lymphocyte activation.

We show that switch-like signaling enables parallel and independent control over response probability and response dynamics: while stimulus area (concentration × duration or antigen quantity × quality) regulates the percentage of cells that respond, the stimulus temporal profile or shape (for example, whether short-strong or low-long or antigen quality/quantity ratio) determines the response timing and gene expression phenotype in responding cells. Dose and duration sensing may be beneficial in different contexts. Dose information is encoded in the delay timing and heterogeneity of NF-κB response. On the other hand, modulating duration on the sub-minute timescale does not regulate response dynamics. Indeed, achieving control of percentage active in a population without introducing heterogeneity requires modulating duration of a high-dose input (*Figure 4*). It was shown that signaling dynamics mediates transfer of input dose information (*Selimkhanov et al., 2014*). We find that while dose information is transmitted through dynamics of NF-κB activation, on short (minute) time scales duration, information is lost in the single-cell response but retained in the population response (fraction of activated cells).

Between the innate immune signals TNF and LPS, we found that LPS exhibits greater ultrasensitivity (a steeper stimulus-response curve) and more pronounced activation delay than TNF. Both of these features are explained by higher coopertivity in IKK activation for LPS than for TNF (*Figure 3—figure supplement 2B*). Distinct higher order adapter protein architectures may activate IKK with different effective coopertivities (*Kazmierczak and Lipniacki, 2010*). We note that the LPS-signaling response in macrophages may differ from that in 3T3 cells, including effects due to stronger auto and paracrine TNF signaling.

While TNF signaling activates formation of a filamentous amyloid complex involving RIP1 and RIP3 kinases, LPS signaling is mediated through helical assembly of the Myddosome complex (*Lin et al., 2010*), which interfaces with a TRAF6 lattice structure to activate IKK (*Yin et al., 2009*). Heterogeneity in switching threshold between cells may arise from cell-to-cell expression differences in signalosome components such as RIP1/3, MyD88, IRAK2/4, and TRAF6, leading to altered kinetics of signalosome assembly and IKK activation. Because the IKK hub mediates NF-κB responses for a multitude of input types and coordinates cross-talk with other signaling pathways, understanding how different signalosome architectures induce specific responses paves the way to interventions directed at switch-like signaling to modulate population and individual cell dynamics towards therapeutic outcomes (*Negro et al., 2008*; *Behar et al., 2013*). In this study, we have shown that the switch-like character of NF-κB activation enables orthogonal control over two critical aspects of the response—probability of activation (fraction of active cells) and the heterogeneity of response—through the integral and temporal profile of the input. Secretion of signaling molecules often occurs in discrete or quantized way in the form of secretory bursts, and particularly in the case of short range paracrine signaling, cells may produce brief but intense secretion to achieve, for example, low probability but high predictability responses (non-heterogeneous dynamics). Overall, these results expand the repertoire of functions for digital signaling beyond increasing robustness to also facilitate multidimensional phenotype control based on temporal information in input signals.

## Materials and methods

### Cell lines

We used p65-knockout 3T3 fibroblasts (courtesy Markus Covert) modified using lentiviral vectors to express p65-DsRed under its endogenous promoter along with an H2B-GFP nuclear reporter, as described previously (*Lee et al., 2009*). The cell line was clonally derived to express at p65-DsRed at lowest detectable level to preserve near endogenous expression.

### Automated microfluidic cell culture system

Automated microfluidic cell culture was performed as previously described (*Gómez-Sjöberg et al., 2007*; *Tay et al., 2010*; *Kellogg et al., 2014*). Briefly, microfluidic chambers were fibronectin treated and seeded with cells at approximately 200 cells/chamber. Cells were allowed to grow for 1 day with periodic media replenishment until 80% confluence. To stimulate cells, media equilibrated to 5% $CO_2$ and containing the desired LPS amount was delivered to chambers, leading to a step increase in LPS concentration. All LPS doses were tested in parallel in a single chip. To produce LPS and TNF pulses, chambers were washed with media after incubation with ligand for the desired duration. Stimulations were applied in duplicate chambers on the chip. Following stimulation, chambers were sealed and imaged at 5- to 6-min intervals.

## Image acquisition and data analysis

DsRed and GFP channels were acquired using a Leica Microsystems (Wetzlar, Germany) DMI6000B widefield microscope at 20× magnification with a Retiga-SRV CCD camera (QImaging - Surrey, BC, Canada) using Leica L5 and Y3 filters to acquire GFP and DsRED signals, respectively, and a Leica EL6000 mercury metal halide light source. One or two images were acquired per chamber and stitched if required using ImageJ (Pairwise stitching plugin). CellProfiler software (www.cellprofiler.org) and custom Matlab software was used to automatically track cells and quantify NF-κB translocation, and automated results were manually compared with images to ensure accuracy prior to further analysis. Mitotic cells were excluded from analysis. NF-κB activation was quantified as mean nuclear fluorescence intensity normalized by mean cytoplasm intensity. Area of the first peak was integrated after baseline correction from the time of LPS stimulation to the first minimum for each cell using Matlab function *trapz*. For peak analysis, data were smoothed (Matlab function *smooth*) followed by peak detection (Matlab function *mspeaks*) to extract NF-κB peak properties (intensity, area, delay) with manual verification using a custom interface in Matlab. Statistical analysis of NF-κB peak amplitude and timing data was performed by unpaired two-sample T-test (Matlab function *ttest2*).

## Gene expression

Cells were seeded at 10,000 cells/well in a 96-well plate and left to attach overnight before stimulation with LPS (1–500 ng/ml). Cells were stimulated with LPS and then lysed, the RNA reverse-transcribed and cDNA pre-amplified (specific target amplification with the set of 24 primers) using the One-Step RT-PCR kit from Invitrogen (San Diego, CA, United States). Quantitative PCR with technical duplicates was carried out on 48.48 dynamic arrays from Fluidigm according to manufacturer instructions, and expression was normalized to GAPDH.

## Additional information

### Funding

| Funder | Grant reference | Author |
| --- | --- | --- |
| European Research Council (ERC) | Starting Grant (SingleCellDynamics) | Ryan A Kellogg, Savaş Tay |
| Schweizerischer Nationalfonds zur Förderung der Wissenschaftlichen Forschung (SNF) | National Centres of Competence in Research (NCCRs) #NCH1668 - Molecular Systems Engineering | Ryan A Kellogg, Savaş Tay |
| Narodowe Centrum Nauki (NCN) | 2011/03/B/NZ2/00281 | Tomasz Lipniacki |

The funders had no role in study design, data collection and interpretation, or the decision to submit the work for publication.

### Author contributions

RAK, ST, Conception and design, Acquisition of data, Analysis and interpretation of data, Drafting or revising the article; CT, TL, Analysis and interpretation of data, Drafting or revising the article; SRQ, Acquisition of data, Contributed unpublished essential data or reagents

### Author ORCIDs

Ryan A Kellogg, http://orcid.org/0000-0002-5591-9466
Chengzhe Tian, http://orcid.org/0000-0002-2269-1979

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

## Appendix 1

## Supplementary mathematical methods

This appendix contains the supplementary information for the mathematical model in the main paper. Firstly, we summarize existing experimental findings about the LPS-mediated NF-κB signaling pathway. Then, we introduce assumptions to simplify the pathway and construct the mathematical model. Finally, we describe the routine to fit the parameters and provide the values used in the model.

## Experimental findings of the LPS-mediated NF-κB pathway

The LPS-mediated NF-κB pathway has been extensively studied experimentally, and **Appendix figure 1** provides a schematic view of the current understanding. The pathway consists of two branches, namely the TLR4-MyD88-dependent branch and the TLR4-TRIF-dependent branch. Both branches converge at IKK, where they activate IKK and trigger the release of NF-κB molecules. The pathway downstream of the IKK activation (core NF-κB module) is well-studied and has been modeled extensively (a review is available in **Cheong et al. (2008)** and our previous work is **Tay et al. (2010)**). Hence, we focus on the upstream part, or how the cell recognizes LPS molecules and activates IKK.

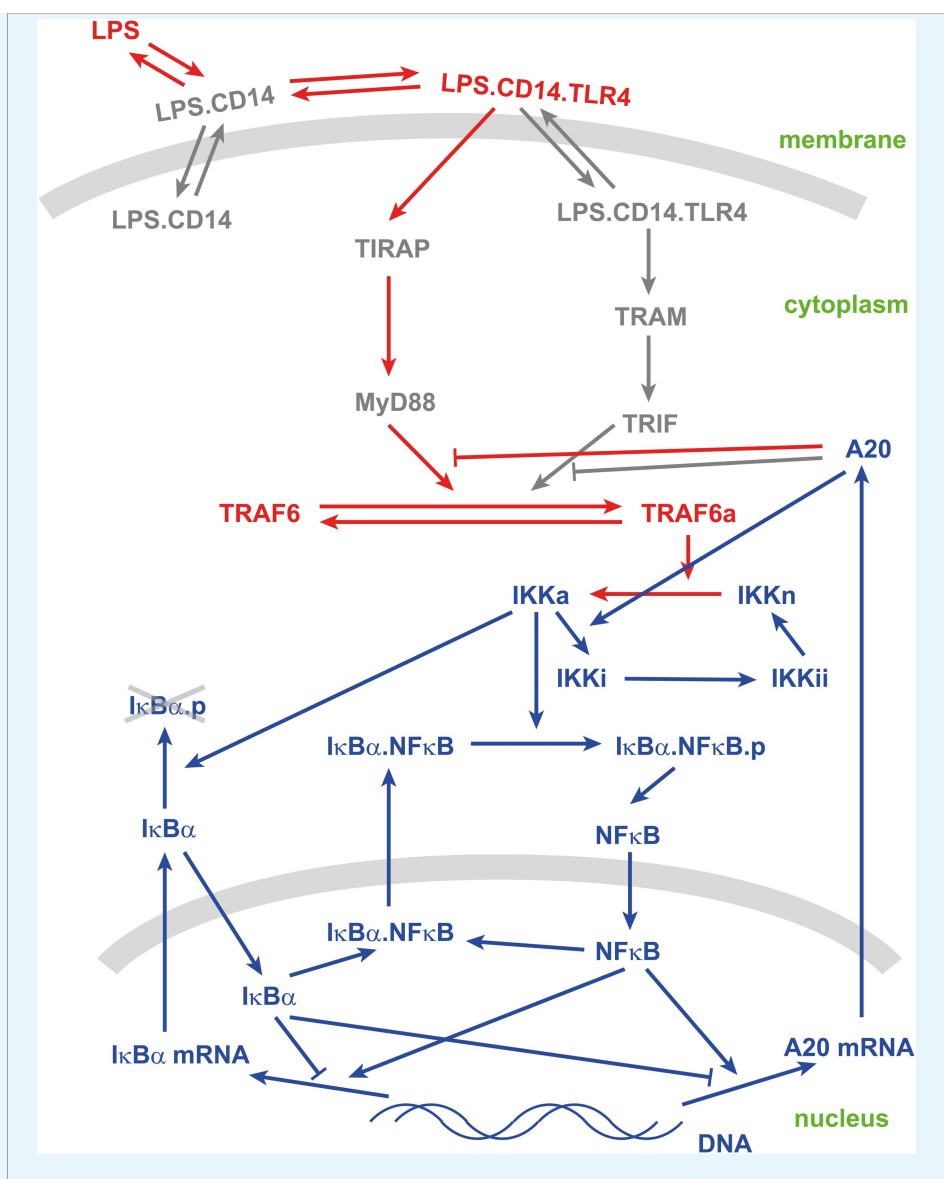

**Appendix figure 1**. Schematic view of the LPS-mediated NF-κB pathway. Text: species; arrows: reactions. Blue: existing species and reactions in the model in *Tay et al. (2010)*; Red: newly introduced species and reactions; Gray: neglected species and reactions.

## TLR4-MyD88-dependent branch

This branch is well-understood and has been systematically reviewed in *Lu et al. (2008)*, *Newton and Dixit (2012)*. LPS molecules first bind CD14 on the membrane, then bind the TLR4 receptors to form complexes. The LPS.CD14.TLR4 complexes recruit TIRAP, MyD88, and other factors to activate TRAF6, which is in the form of clusters (*Yin et al., 2009*). The activated TRAF6 molecules then phosphorylate TAK1, which will activate IKK and trigger the response of NF-κB. It has been shown that the TLR4-MyD88-dependent branch induces the early response.

## TLR4-TRIF-dependent branch

CD14 has been reported to facilitate the endocytosis of TLR4 receptors (*Zanoni et al., 2011*), and the LPS.CD14.TLR4 complexes can also trigger the innate immunity in the cytoplasm. The current reviews about the TLR4-TRIF-dependent branch are conflicting. Lu et al. and Newton et al. claim that the LPS.CD14.TLR4 complexes first recruit TRIF through the adapter TRAM.

Then, RIP1 is activated and TAK1 is phosphorylated. Finally, the phosphorylated TAK1 molecules activate IKK and trigger the response of NF-κB (**Lu et al., 2008**; **Newton and Dixit, 2012**). O'Neill et al. states that the RIP1 molecules first get activated by the LPS.CD14.TLR4 complexes, then TRAF6 is activated by RIP1, and finally TAK1 is phosphorylated by TRAF6 (**O'Neill et al., 2013**). Meanwhile, Kawai et al. claims that both RIP1 and TRAF6 can be activated by the LPS.CD14.TLR4 complexes and can both phosphorylate TAK1 (**Kawai and Akira, 2010**). The TLR4-TRIF-dependent branch is shown to induce the late response.

In addition to these three views, Covert et al. proposed that the TLR4-TRIF-dependent branch activates NF-κB indirectly (**Covert et al., 2005**; **Lee et al., 2009**). The claim is that the TLR4-TRIF-dependent branch first activates the production of TNFα, then TNFα activates the response of NF-κB through the well-known TNFα signaling pathway. The time required for activating the TNFα production is the cause for the delay in the NF-κB response. However, we do not observe such phenomenon in our experiments and therefore ignore this mechanism when constructing the mathematical model.

## Model construction and simplification

In this section, we describe the procedure for constructing the mathematical model based on the existing experimental knowledge. We construct the model based on our previous TNFα model in **Tay et al. (2010)** whose performance has been constantly validated by the experiments in our group. In order to provide with a unified framework, we model the reactions in the LPS-mediated NF-κB pathway in the same fashion: the equations for the core NF-κB module are identical to the ones in the TNFα model, and the equations for the pathways upstream of the IKK activation are described as follows.

### TLR4-MyD88-dependent branch

We model the concentrations of LPS, CD14, LPS.CD14 complex, LPS.CD14.TLR4 complex, and TRAF6, and the reactions in this branch are shown in **Appendix table 1**. In the model, we attribute the cell–cell variability upstream of the IKK activation to the receptor-level, as we assume the abundances of CD14 and TLR4 follow lognormal distributions. We use mass action laws to model the association and dissociation of the ligands and receptors (LPS, CD14, and TLR4). The internalization and the externalization processes are modeled by mass action laws as well. We assume the activity of MyD88 is proportional to the concentration of LPS.CD14.TLR4 complex, therefore, the activation rate of TRAF6 is proportional to the product of the concentrations of LPS.CD14.TLR4 and TRAF6. We also use a Michaelis-Menten term to model the inhibitory effect of A20, as in the TNFα model. The inactivation of TRAF6a is modeled by a mass action law. Finally, we use a Hill term to model the clustering of activated TRAF6, with $n$ representing the cluster sizes and $K_{IKK}$ representing a critical level, in order to illustrate the effect of clustering explicitly. The activation rate of IKK is then proportional to the product of the Hill term and the concentration of IKKn.

**Appendix table 1**. The reactions in the TLR4-MyD88-dependent branch

| Reaction/Description | Rate laws |
|---|---|
| $LPS \rightarrow \varnothing$ | $l_d[LPS]$ |
| Degradation of LPS | |
| $LPS + CD14 \rightarrow LPS.CD14$ | $l_{b,CD14}[LPS][CD14]$ |
| Association of LPS and CD14 | |
| $LPS.CD14 \rightarrow LPS + CD14$ | $l_{f,CD14}[LPS.CD14]$ |
| Dissociation of LPS.CD14 | |
| $LPS.CD14 \rightarrow LPS.CD14.int$ | $l_{in,CD14}[LPS.CD14]$ |
| Internalization of LPS.CD14 | |

*Appendix table 1. Continued on next page*

*Appendix table 1. Continued*

| Reaction/Description | Rate laws |
|---|---|
| LPS.CD14.int → LPS.CD14 | $l_{out,CD14}[LPS.CD14.int]$ |
| Externalization of LPS.CD14.int | |
| LPS.CD14 + TLR4 → LPS.CD14.TLR4 | $l_b[LPS.CD14][TLR4]$ |
| Association of LPS.CD14 and TLR4 | |
| LPS.CD14.TLR4 → LPS.CD14 + TLR4 | $l_f[LPS.CD14.TLR4]$ |
| Dissociation of LPS.CD14.TLR4 | |
| TRAF6 → TRAF6a | $l_a[LPS.CD14.TLR4][TRAF6]\frac{k_{M,A20}}{k_{M,A20}+[A20]}$ |
| Activation of TRAF6 by LPS.CD14.TLR4 | |
| TRAF6a → TRAF6 | $l_i[TRAF6a]$ |
| Inactivation of TRAF6 | |
| $IKK_n$ → $IKK_a$ | $c_1\frac{[TRAF6a]^n}{[TRAF6a]^n+K_{IKKK}^n}[IKKn]$ |
| Activation of IKK by TRAF6a | |

## TLR4-TRIF-dependent branch

As mentioned previously, there are at least three conflicting views about this branch. Here, we do not distinguish RIP1 from TRAF6 and assume internalized LPS.CD14.TLR4 complexes activate TRAF6 directly, since this assumption can be compensated by the parameter values. We model the reactions in this branch in the same fashion as the TLR4-MyD88-dependent branch, and the equations (in addition to the ones in the TLR4-MyD88-dependent branch) are listed in **Appendix table 2**. Here, we assume that the Michaelis-Menten constant for the repressed activation of TRAF6 by the internalized LPS.CD14.TLR4 is the same as the one in the TLR4-MyD88-dependent branch. The underlying reason is that A20 functions by removing the ubiquitin chains, and it can be assumed irrelevant to the sources of activation.

**Appendix table 2**. The additional reactions in the TLR4-TRIF-dependent branch

| Reaction/Description | Rate laws |
|---|---|
| LPS.CD14.TLR4 → LPS.CD14.TLR4.int | $l_{in}[LPS.CD14.TLR4]$ |
| Internalization of LPS.CD14.TLR4 | |
| LPS.CD14.TLR4.int → LPS.CD14.TLR4 | $l_{out}[LPS.CD14.TLR4.int]$ |
| Externalization of LPS.CD14.TLR4.int | |
| TRAF6 → TRAF6a | $l_{a,int}[LPS.CD14.TLR4.int][TRAF6]\frac{k_{M,A20}}{k_{M,A20}+[A20]}$ |
| Activation of TRAF6 by LPS.CD14.TLR4.int | |

## Model simplification

In principle, the model has the potential to describe the dynamical behavior of the LPS-mediated NF-κB pathway accurately. However, the model has approximately 20 free parameters, and it is practically impossible to perform a large-scale sampling in the parameter space. Hence, we introduce two assumptions to simplify the model.

### Removing CD14 from the model

In humans, the quantity of CD14 is approximately $6.2 \times 10^5$ molecules per cell, and the quantity of TLR4 is approximately $10^2$–$10^3$ molecules per cell (**Kitchens and Munford, 1998**), (**Pawelczyk et al., 2010**). Although to the authors' knowledge, there is no extensive measurement of the numbers in mouse NIH3T3 cells, we make the assumption that CD14 is excessive and the variability of CD14 abundance is not the main source of cell–cell variability.

Hence, we remove CD14 from the model and assume LPS directly binds TLR4 receptors. We also assume that TLR4 abundance accounts for all the cell–cell variability upstream of the IKK activation. It is worth mentioning that the effect of CD14 can be compensated by the binding and unbinding rates and the effect of the internalization of LPS.CD14 can be compensated by a higher degradation rate of LPS. Therefore, this assumption will not restrict the performance of the model.

### Removing the TLR4-TRIF-dependent branch

In our model, the TLR4-MyD88-dependent branch and the TLR4-TRIF-dependent branch are identical in structure. Since the two branches have the same function of activating TRAF6, and we focus on the overall dynamical behavior of the LPS-mediated NF-κB pathway, it is not necessary to distinguish the two branches explicitly. To reduce the complexity the model, we assume the LPS.TLR4 complexes do not internalize and remove the TLR4-TRIF-dependent branch. In addition, except for the constantly perfused LPS experiments, we mostly focus on the first peak of the NF-κB response and such peak is triggered by the TLR4-MyD88-dependent branch. So only modeling this branch will not sacrifice the quality of the model with respect to predictive power in our experimental settings.

### Final model

With the two simplifications, we reduce the complexity of the full model and the number of free parameters is reduced to around 10. The structure of the model is highlighted in red and blue in *Appendix figure 1*. The newly introduced reactions (in red) are listed in *Appendix table 3*. The final model significantly simplifies the pathway upstream of the IKK activation, and this simplification can be justified from the perspective of information flow. The two steps in the model, namely the binding/unbinding of ligands and receptors and the activation/inactivation of TRAF6, correspond to two important processing of the information flow: the first step encodes the cell–cell variability into the information flow, and the second step allows the analog information flow to trigger a digital decision. The reactions that are not covered in the final model process the information flow in a less important manner, so the final model can be justified within our scope of study.

**Appendix table 3**. The reactions upstream of IKK activation in the final model

| Reaction/Description | Rate laws |
|---|---|
| $LPS \rightarrow \varnothing$ | $l_d[LPS]$ |
| Degradation of LPS | |
| $LPS + TLR4 \rightarrow LPS.TLR4$ | $l_b[LPS][TLR4]$ |
| Association of LPS and TLR4 | |
| $LPS.TLR4 \rightarrow LPS + TLR4$ | $l_f[LPS.TLR4]$ |
| Dissociation of LPS and TLR4 | |
| $TRAF6 \rightarrow TRAF6a$ | $l_a[LPS.TLR4][TRAF6]\frac{k_{M,A20}}{k_{M,A20} + [A20]}$ |
| Activation of TRAF6 by LPS.TLR4 | |
| $TRAF6a \rightarrow TRAF6$ | $l_i[TRAF6a]$ |
| Inactivation of TRAF6a | |
| $IKKn \rightarrow IKKa$ | $c_1\frac{[TRAF6a]^n}{[TRAF6a]^n + K_{IKKK}^n}[IKKn]$ |
| Activation of IKKn by TRAF6a | |

## Parameter estimation

Despite simplification, the complexity of the model is high and the model is stochastic in its nature. Hence, we use a combined approach to determine the parameter values. Firstly, we fix the distribution of the number of TLR4 receptors and convert the measurement of LPS from

concentration to molecular number. Then, we iterate over automatic and manual sampling of the parameter space to determine the final values.

## Description of approaches

### Distribution of TLR4 abundance

As with the TNFα model in *Tay et al. (2010)*, the total number of TLR4 receptors on the cell membrane follows a lognormal distribution. To our knowledge, there is no accurate measurement on mouse NIH3T3 cells, but the intuition is that the number is in the magnitude of $10^3$. Here, we assume that the parameters for the lognormal distributions are $\mu = 8.0$ and $\sigma = 0.8$. Therefore, the expected number of TLR4 receptors is $4.1 \times 10^3$, the standard deviation is $3.9 \times 10^3$, the median is $3.0 \times 10^3$, and the mode is $1.6 \times 10^3$. An illustration of the distribution is available in *Appendix figure 2*.

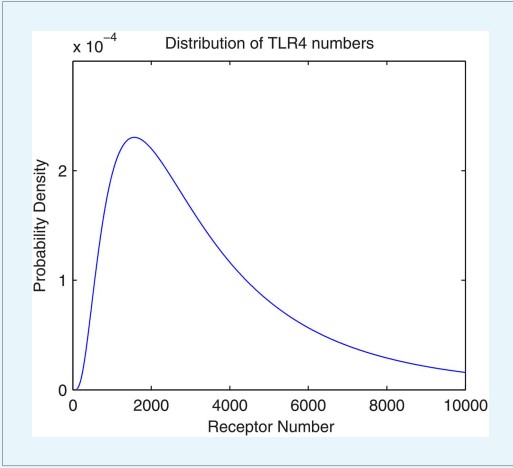

**Appendix figure 2**. The distribution of the number of TLR4 receptors in the model. The number follows lognormal distribution with parameters $\mu = 8.0$ and $\sigma = 0.8$.

### Converting LPS concentrations to molecular numbers

The experiments use concentrations to characterize the abundance of LPS while the model uses molecular numbers. Here, we perform a conversion to unify the units used in the model. The molecular weight of LPS is approximately 10 kDa (*Sigma-Aldrich, 2008*) and the size of each chamber in the microfluidic device is 1.12 mm × 0.9 mm × 0.040 mm (*Gómez-Sjöberg et al., 2007*). Since there are around 100 cells in each chamber, for 1 ng/ml LPS, the average number of ligands each cell can interact with is $2.4 \times 10^4$. Therefore, we can produce a conversion table for LPS, as shown in *Appendix table 4*.

**Appendix table 4**. The conversion table of LPS between concentrations and molecular numbers

| Concentrations ($/\mathrm{ng \cdot ml^{-1}}$) | Molecular numbers |
| --- | --- |
| 500 | $1.2 \times 10^7$ |
| 100 | $2.4 \times 10^6$ |
| 50 | $1.2 \times 10^6$ |
| 10 | $2.4 \times 10^5$ |
| 5 | $1.2 \times 10^5$ |
| 1 | $2.4 \times 10^4$ |
| 0.5 | $1.2 \times 10^4$ |
| 0.25 | $6.0 \times 10^3$ |

## Iterative approach of parameter estimation

The automatic and manual methods share the same objective. For each set of parameter values, we perform the simulation 200 times, each time with newly sampled numbers of TLR4 and NF-κB from the two distributions. One cell is considered activated if the amplitude of the first peak is at least 10% of the total number of NF-κB. Then, the fractions of activated cells for all the input conditions are calculated and the set of parameter values is appropriate if the L2-norm error between the estimated fractions and the experimentally measured fractions is small.

The automatic method first samples the values of the parameter equidistantly in the log-scale. Then, the script calculates the fractions of activation for all the sets of parameter values. Finally, the set with the least L2-norm error is selected. This approach is objective, but a large amount of computation power is required. Meanwhile, the manual approach is to change the parameter value manually and this method is good for obtaining intuition about the model. Here, we integrate the two methods by iterating them, and the parameter values are able to be determined in a reasonable time and accuracy.

## Parameter values

The estimated parameter values for the model are listed in **Appendix table 5**, and the remaining parameters take the values in the TNF$\alpha$ model in **Tay et al. (2010)**.

**Appendix table 5**. The parameter values used in the model

| Parameters | Biological meanings | Values | Remarks |
|---|---|---|---|
| $l_d$ | Degradation rate of LPS | $5 \times 10^{-4}$ | Fitted |
| $l_b$ | Binding rate of LPS and TLR4 | $5 \times 10^{-9}$ | Fitted |
| $l_f$ | Unbinding rate of LPS.TLR4 | $4.5 \times 10^{-4}$ | Fitted |
| $l_a$ | Activation rate of TRAF6 | $1 \times 10^{-7}$ | Fitted |
| $k_{M,A20}$ | Constant for repressed TRAF6 activation | $1 \times 10^5$ | Assume same as **Tay et al. (2010)** |
| $l_i$ | Inactivation rate of TRAF6a | $1 \times 10^{-2}$ | Assume same as **Tay et al. (2010)** |
| $c_1$ | Activation rate of IKK | $2 \times 10^{-2}$ | Fitted |
| $K_{IKKK}$ | Dissociation constant for IKK activation | $3.5 \times 10^3$ | Fitted |
| $n$ | Hill coefficient | 4 | Fitted, with support from **Yin et al. (2009)** |
| $M_{IKKK}$ | Total number of TRAF6 | $10^5$ | Assume same as **Tay et al. (2010)** |

To illustrate the effect of clustering, we also construct a model with Hill coefficient $n = 1$. We perform the same procedure to estimate the parameter values, and the new values for $c_1$ and $K_{IKKK}$ are $2 \times 10^{-4}$ and $7.5 \times 10^3$, respectively. The remaining parameters are not modified.

# Dose-duration relationship

Activation of cells by LPS treatment requires a high dose and a long duration. In this section, we analyze the relationship between the fraction of activated cells (population response) and the dose/duration of the LPS input.

## Analytical expression of the fraction of activation

For simplicity, consider one elementary reversible reaction

$$A + B \underset{k_+}{\overset{k_-}{\rightleftharpoons}} AB,$$

where the input signal $A$ binds the receptor $B$ to form a complex $AB$ with a forward kinetic constant $k_+$ and $AB$ dissociates with a backward kinetic constant $k_-$. The amount of receptors is conserved ($[B] + [AB] = B_0$ is a constant) and $A$ is an arbitrary non-negative function of time $t$ to illustrate a general input profile. The concentration of $AB$ follows

$$\frac{\mathrm{d}[AB]}{\mathrm{d}t} = k_+[A](B_0 - [AB]) - k_-[AB],$$

and the closed-form solution of $[AB]$ is

$$[AB] = \exp\left\{-\int_0^t (k_+[A] + k_-)\mathrm{d}\tau\right\}\left\{\int_0^t e^{\int_0^\tau (k_+[A] + k_-)\mathrm{d}\tau'} k_+[A]B_0\mathrm{d}\tau\right\}, \qquad (1)$$

assuming the initial condition $[B](0) = B_0$. The maximum value of $[AB]$ should be greater than the threshold value $[AB]_{th}$ to active the cells.

When the dose of the input $A$ is high such that the backward reaction can be neglected, by defining the total area under the input from time 0 to $t$ as $S(t) = \int_0^t [A]\mathrm{d}\tau$, **Equation 1** becomes

$$[AB] = B_0\left(1 - e^{-k_+S}\right).$$

We assume that $B_0$ follows a probability distribution to illustrate the receptor-level variability, the fraction of activation given an input area $S$ is

$$P\left([AB] \geq [AB]_{th}\right) = P\left(B_0 \geq \frac{[AB]_{th}}{1 - e^{-k_+S}}\right). \qquad (2)$$

Since the forward kinetic constant $k_+$ and the receptor-level variability is known, the fraction of activation therefore is only dependent on the area under the input $S$.

## The caveat of the analysis

**Equation 2** holds for general signaling systems with arbitrary input signals given that the backward reaction can be neglected. If the input is weak, the maximal concentration of $AB$ is

$$\max[AB] = \frac{k_+[A]}{k_+[A] + k_-}B_0.$$

For a cell with a given amount of receptors ($B_0$), if $\max[AB] < [AB]_{th}$, the cell is not able to activate regardless of the input area. Similarly, given a population of cells, the maximal fraction of activation is

$$\max P\left([AB] \geq [AB]_{th}\right) = P\left(B_0 \geq [AB]_{th}\frac{k_+[A] + k_-}{k_+[A]}\right).$$

A longer duration of input will not increase the fraction of activation over this value. Therefore, the 'high dose' requirement illustrates a caveat of this analysis.

It is worth mentioning that the term 'high dose' here indicates that a dose where the backward reaction is neglected, while 'high dose' in the main paper indicates a dose where nearly all the cells are activated. The analysis is also applicable to many doses where only the minority of cells are activated (for example **Figure 6B**).

