## [Decision Letter]

Thank you for submitting your work entitled “Digital signaling decouples activation probability and population heterogeneity” for peer review at *eLife*. Your submission has been favorably evaluated by Naama Barkai (Senior editor) and three reviewers, one of whom is a member of our Board of Reviewing Editors.

The reviewers have discussed the reviews with one another and the Reviewing editor has drafted this decision to help you prepare a revised submission.

This is an interesting study of digital signaling dynamics using LPS-stimulated NF-kB activation in 3T3 cells as a model system. Microfluidic analysis is used to follow the behavior of single cells. The authors report the analysis of a dose response (under non-pulse conditions) and a pulse-length study with a high LPS dose together with simulation studies based on a mathematical model. The authors conclude that the fractional digital response to LPS is increased with both dose and pulse length. Please address the following points in a revised submission:

A) The experimental data requires statistical analysis. Error bars are missing in some Figure panels (or are present, but not defined in some other panels) and the number of biological replicates is not stated. Moreover, conclusions based on the experimental data are not supported by statistical tests.

B) The term “full response” should be to be clarified. A “full response” implies that most, if not all, NF-κB translocates to the nucleus (as in high dose stimulation), yet Figure 2 (0.25ng/ml LPS) does not appear to show this. In addition, the authors state that the Nuc-NF-κB is constant on average, yet the single-cell traces clearly gives the impression that the amplitude increases with respect to the concentration. Figure 3 (model) predicts quite striking amplitude changes as the dose is varied. Moreover, the simulation (Figure 3—figure supplement 1) uses different scales for different treatments (showing the amplitude decreasing with lowering the dose). In addition, the pulse-length study with TNF (Figure 5—figure supplement 1) appears to show a graded response, but this is not represented in Figure 5—figure supplement 1).

C) The conclusion of a Hill coefficient of 2 (Figure 2) is problematic because of the absence of data between 1 and 5ng/ml. Moreover, if the Hill coefficient is 2, why is a Hill coefficient of 4 used in the model?

D) The criteria for inclusion/exclusion of NF-kB pathway characteristics in the model are unclear. Is variable TLR4/CD14 expression the mechanism of NF-κB dynamics variability? Is MyD88 the mechanism of non-linearity? What justifies the focus on these features of the NF-kB pathway?

E) The model simulation data are interesting, but the usefulness of the model is to generate hypotheses concerning signaling network behavior. The authors need to experimentally test a model-based prediction. In this case, the model is used to predict the response to different pulse length variations for intermediate LPS doses (Figure 6). However, the experimental data shown is limited to a dose response (under non-pulse conditions) and a pulse-length study with a high LPS dose.

---

## [Author Response]

A) The experimental data requires statistical analysis. Error bars are missing in some Figure panels (or are present, but not defined in some other panels) and the number of biological replicates is not stated. Moreover, conclusions based on the experimental data are not supported by statistical tests.

We performed statistical analysis of all amplitude and timing data and included this in as additional supplementary figures associated with Figures 2 and 5. These new figures are Figure 2—figure supplement 2, Figure 5—figure supplement 2, and Figure 5—figure supplement 3. We now refer to this analysis in the claims in the main text (subsections “NF-κB switch dynamics distinguish pathogen (LPS) and host (TNF) signals”, “Response timing and single-cell heterogeneity depends on stimulus intensity” and “Input duration controls activation probability independent of response heterogeneity”). Error bars are included in all places applicable and we clarify the meaning of error bars in the figure captions. Data are plotted as median with interquartile range. The manuscript text is updated to reflect the number of biological replicates, two duplicate microfluidic chambers per condition (subsections “NF-κB switch dynamics distinguish pathogen (LPS) and host (TNF) signals” and “Automated microfluidic cell culture system”).

*B) The term “full response” should be to be clarified. A “full response” implies that most, if not all, NF-κB translocates to the nucleus (as in high dose stimulation), yet*
Figure 2
*(0.25ng/ml LPS) does not appear to show this. In addition, the authors state that the Nuc-NF-κB is constant on average, yet the single-cell traces clearly gives the impression that the amplitude increases with respect to the concentration.*
Figure 3
*(model) predicts quite striking amplitude changes as the dose is varied. Moreover, the simulation (*Figure 3—figure supplement 1*) uses different scales for different treatments (showing the amplitude decreasing with lowering the dose). In addition, the pulse-length study with TNF (*Figure 5—figure supplement 1*) appears to show a graded response, but this is not represented in*
Figure 5—figure supplement 1*).*

We have revised the text to clarify these points. We do observe responses at lowest doses where nearly all NF-κB enters the nucleus (see Figure 2). Amplitude is highly variable across doses, and while median amplitude increases gradually with dose the change is statistically less significant than changes in response time (Figure 2—figure supplement 2) (subsection “NF-κB switch dynamics distinguish pathogen (LPS) and host (TNF) signals”). We acknowledge greater amplitude dependence on dose in the model (subsection “Cooperative IKK activation underlies dose-dependent response delay”) and add notes to indicate the changing y-scale in Figure 3—figure supplement 1 and Figure 4—figure supplement 1. Figure 5—figure supplement 1 represents analysis of the traces presented in Figure 5—figure supplement 1. Statistical analysis shows overall that amplitude does modulate significantly with duration change (Figure 5—figure supplement 3; subsection “Input duration controls activation probability independent of response 151 heterogeneity”).

*C) The conclusion of a Hill coefficient of 2 (*Figure 2*) is problematic because of the absence of data between 1 and 5ng/ml. Moreover, if the Hill coefficient is 2, why is a Hill coefficient of 4 used in the model?*

The experimental hill coefficient refers to the steepness in increase in fraction of activating cells with increasing input level. We incorporated analysis of 3ng/ml LPS into Figure 2, which activated 40.6% of the population. Refitting the Hill curve led to a steepness coefficient of 2.3, which we updated in the main text (subsection “NF-κB switch dynamics distinguish pathogen (LPS) and host (TNF) signals”).

The comment about hill coefficient of 4 in the model (subsection “Cooperative IKK activation underlies dose-dependent response delay”, last paragraph) refers to the cooperativity in IKK activation by TRAF6 (distinct from the slope in the fraction of activation as a function of dose). We now term this value cooperativity coefficient and add a note to avoid this confusion.

D) The criteria for inclusion/exclusion of NF-kB pathway characteristics in the model are unclear. Is variable TLR4/CD14 expression the mechanism of NF-κB dynamics variability? Is MyD88 the mechanism of non-linearity? What justifies the focus on these features of the NF-kB pathway?

Inclusion/exclusion of NF-κB pathway characteristics in the model is based on the available information about LPS-NF-κB signaling, maintaining model simplicity, and following methodology consistent with currently accepted NF-κB models. The mathematical supplement [Sec s1 s2] contain substantial discussion of our process and decisions in constructing the model.

To explain observed heterogeneity in cell responses it is enough to assume that cells have different sensitivities to the signal, which can result from broad distribution of receptors. Receptor level variation is a major source of heterogeneity between cells, though we acknowledge that other sources of heterogeneity could contribute (subsection “Cooperative IKK activation underlies dose-dependent response delay”). The nonlinearity results from clustering and oligomerization of receptors and adapter proteins including MyD88 and TRAF6. Multiple studies support that clustering mediates digital activation in NF-κB signaling (discussed in the aforementioned subsection) though we discuss other cases where additional factors may contribute (Introduction, first paragraph). We have updated the supplemental mathematical materials for clarity and with additional details of the methodology and justification in building the model.

*E) The model simulation data are interesting, but the usefulness of the model is to generate hypotheses concerning signaling network behavior. The authors need to experimentally test a model-based prediction. In this case, the model is used to predict the response to different pulse length variations for intermediate LPS doses (*Figure 6*). However, the experimental data shown is limited to a dose response (under non-pulse conditions) and a pulse-length study with a high LPS dose.*

We use the model to generate the hypothesis that integral of the input determines the fraction of activating cells, which is supported by the experimental dose response and high-dose pulse data. We now test this prediction in new experiments further by comparing inputs with different temporal profiles but equal integral (area). We observed 500ng/ml LPS pulsed for 10 s activated approximately half the population (Figure 5). Therefore we tested two additional input profiles having the same area (5000 ngml^-1^/s): 50ng/ml for 100s and 100ng/ml for 50s. In agreement with model prediction, each of these conditions also activates a similar fraction of the population (51% and 54%, respectively). This is discussed in the main text (subsection “Integral of stimulus determines fraction of active cells in the population”, last paragraph) and Figure 7 has been updated with trajectories and images from these experiments.